# Genomic characterization of novel bat kobuviruses in Madagascar: Implications for viral evolution and zoonotic risk

Freddy L. Gonzalez[1,2*], Gwenddolen Kettenburg[1], Hafaliana Christian Ranaivoson[1,3,4,5], Angelo Andrianiaina[3,5], Santino Andry[6,5], Vololoniaina Raharinosy[4], Tsiry Hasina Randriambolamanantsoa[4], Vincent Lacoste[4], Philippe Dussart[4], Jean-Michel Héraud[4¤], Cara E. Brook[1]

**1** Department of Ecology and Evolution, University of Chicago, Illinois, United States of America, **2** Department of Ecology and Evolutionary Biology, Yale University, Connecticut, United States of America, **3** Department of Zoology and Animal Biodiversity, University of Antananarivo, Madagascar, **4** Virology Unit, Institut Pasteur de Madagascar, Anatananarivo, Madagascar, **5** Ekipa Fanihy Association, University of Antananarivo, Madagascar, **6** Department of Entomology, University of Antananarivo, Madagascar

¤ Current address: World Health Organization, Geneva, Switzerland
* freddy.gonzalez@yale.edu (FLG)

## Abstract

Kobuviruses (family *Picornaviridae*, genus *Kobuvirus*) are enteric viruses that infect a wide range of both human and animal hosts. Much of the evolutionary history of kobuviruses remains elusive, largely due to limited screening in wildlife. Bats have been implicated as major sources of virulent zoonoses, including coronaviruses, henipaviruses, lyssaviruses, and filoviruses, though much of the bat virome still remains uncharacterized. While most bat virus research has historically focused on immediately recognizable zoonotic clades (e.g., SARS-related coronaviruses), a handful of prior reports catalog kobuvirus carriage in bats and posit the role of bats as progenitors of downstream kobuvirus evolution. As part of a multi-year study, we carried out metagenomic Next Generation Sequencing (mNGS) on fecal samples obtained from endemic, wild-caught Madagascar fruit bats to characterize potentially zoonotic viruses circulating within these populations. The wild bats of Madagascar represent diverse Asian and African phylogeographic histories, presenting a unique opportunity for viruses from disparate origins to mix, posing a significant public health threat. Here, we report detection of kobuvirus RNA in Malagasy fruit bats (*Eidolon dupreanum*) and undertake phylogenetic characterization of Malagasy kobuvirus sequences, which nest within the Aichivirus A clade – a kobuvirus clade known to infect a wide range of hosts including humans, rodents, canids, felids, birds, and bats. Given the propensity of kobuviruses for recombination and cross-species transmission, further characterization of this clade is critical for accurate evaluation of future zoonotic threats.

**Data availability statement:** All sixteen annotated contigs were submitted to NCBI and assigned accession numbers OP287812, OR082796, OQ818322, and PV833570-PV833582. Detailed descriptions of analyses are available on our GitHub (https://github.com/brooklabteam/Madagascar-Bat-Kobuvirus).

**Funding:** Funding: This work was funded by the National Institutes of Health (1R01AI129822-01 grant to J-MH, PD, and CEB and 5DP2AI171120 grant to CEB), DARPA (PREEMPT Program Cooperative Agreement no. D18AC00031 to CEB), the Bill and Melinda Gates Foundation (GCE/ID OPP1211841 to CEB and J-MH), the Adolph C. and Mary Sprague Miller Institute for Basic Research in Science (postdoctoral fellowship to CEB), the Branco Weiss Society in Science (fellowship to CEB), the Chan Zuckerberg Biohub, and the University of Chicago PREP program (5R25GM066522 grant, fellowship to FLG). The funders had no role in study design, data collection and analysis, decision to publish, or preparation of the manuscript.

**Competing interests:** No authors have competing interests.

## Introduction

Picornaviruses in the viral family *Picornaviridae* are non-enveloped RNA viruses that infect a wide range of vertebrates, from birds and fish to a variety of mammals, including both humans and bats [1,2]. Famed human picornaviruses include poliovirus (genus: *Enterovirus*), which causes the paralyzing human disease poliomyelitis [3], and the rhinoviruses (also genus: *Enterovirus*) which cause the common cold [4,5]. Arguably the most well-known animal picornavirus is the first described, Foot-and-Mouth-Disease Virus (genus: *Apthovirus*), which causes the agriculturally-devastating disease of the same name in cloven-hoofed animals [6]. To date, bat picornaviruses have been generally overlooked as potential zoonotic pathogens due to the lack of documented zoonotic spillover events in this clade [7]. This oversight has resulted in limited bat picornavirus surveillance which, in turn, hinders efforts to describe their zoonotic potential. In contrast, bat coronaviruses have garnered significant attention due to their established zoonotic potential. Unlike coronaviruses, which exhibit tight coevolutionary relationships with specific bat host species, picornaviruses show low host specificity, as highly similar variants have been detected across a wide array of bat species, suggesting a more generalized host range [2,8]. Further screening for bat picornaviruses in high-risk areas of wildlife-human interaction will provide crucial insights into their evolutionary history and potential for cross-species transmission.

Kobuviruses represent one clade of many recently discovered enteric picornaviruses known to cause severe gastroenteritis in humans and animals [7,9]. As a clade, they are subdivided into genotypes Aichivirus A-F. Genotypes falling under the Aichivirus A classification are hosted by humans, canids, rodents, felids, birds, and bats [10–12], and those within the Aichivirus B-F classification are hosted by cattle, swine, sheep, rabbits, and bats [12–19]. Structurally, kobuviruses are small (~30−32 nm), icosahedral, non-enveloped viruses with a single-stranded positive sense RNA genome of 8.2–8.4 kb in length [19]. They contain only one open reading frame (ORF), which encodes three structural proteins (VP0, VP3, and VP1) and eight nonstructural proteins (L, 2A, 2B, 2C, 3A, 3B, 3C, and 3D) [20]. These genomic and structural features not only play a critical role in the ability of kobuviruses to infect a wide range of hosts but also provide valuable insight into their evolutionary adaptability and potential for zoonotic transmission.

Bats (order Chiroptera) represent one clade of many known mammalian kobuvirus hosts. As the second largest order of mammals, accounting for ~25% of all mammalian species [21], bats are acknowledged reservoir hosts for several highly virulent zoonotic viruses, including rabies and other lyssaviruses, Hendra and Nipah henipaviruses, Ebola and Marburg filoviruses, and SARS and MERS coronaviruses [22–24] — pathogens which they appear to host without experiencing substantial clinical disease [25]. Bats' notable viral tolerance appears to result from pleiotropic impacts of unique anti-inflammatory molecular pathways thought to have originally evolved to mitigate the accumulation of intensive physiological stress accrued during the metabolically costly activity of powered flight, itself

unique to bats among all mammals [25]. Bats also demonstrate uniquely gregarious lifestyles with high population densities, migratory behavior, and long lifespans and boast the largest geographical distribution of all mammals apart from primates [26–28]. Despite bats' unique capacity as virus hosts, studies on bats as reservoirs for kobuviruses remain limited [12,26,29,30]. Prior genomic studies suggest that bat kobuviruses share high genomic identity with rodent kobuviruses [12], indicating a shared ancestral history of likely cross-species transmission. Given this evolutionary history, zoonotic surveillance efforts for kobuviruses should prioritize regions where bat populations overlap with dense human and animal populations.

Madagascar is home to 51 species of bat [31], many of which are endemic and have undergone long evolutionary divergence from sister species in both Africa and Asia [32–35]. Recent evidence identifies Madagascar fruit bats as hosts for numerous circulating viruses [36–40], some of which are potentially zoonotic. Additionally, longitudinal serological surveillance shows that female Malagasy bats exhibit elevated antibody titers during the periods of gestation and reproduction, suggesting that their exposure to viruses such as henipaviruses and filoviruses may be seasonally linked. Routine bat virus surveillance is thus a critical public health priority, particularly in regions where human and bat geographic ranges overlap [41]. With rapidly changing ecology, urbanization, climate change, increased travel, and fragile public health systems, the frequency of bat zoonoses is likely to rise [42]. Despite growing evidence of viral circulation in Madagascar's diverse bat populations, the prevalence and zoonotic potential of many viral taxa—including kobuviruses—remain largely unexplored in these hosts.

Here, we carried out metagenomic Next Generation Sequencing (mNGS) of RNA extracted from fecal and urine samples collected from three endemic Malagasy fruit bat species (*Pteropus rufus, Eidolon dupreanum, Rousettus madagascariensis*). We present the first detection and characterization of any kobuvirus circulating within Malagasy fruit bats and use phylogenetic tools to demonstrate that Malagasy fruit bat kobuviruses share immediate ancestry with previously described members of kobuvirus sub-clade Aichivirus A, a clade that includes several human-infecting species. We aim for our descriptive work to provide a baseline on which future work will build understanding of kobuvirus dynamics within Malagasy wildlife hosts and evaluate their potential capacity for cross-species transmission.

## Materials and methods

### Bat sampling

As part of a multi-year study examining the dynamics of potentially zoonotic viruses in three endemic species of Madagascar fruit bats (*Pteropus rufus, Eidolon dupreanum, Rousettus madagascariensis*), bats were captured in species-specific roost sites in the Districts of Moramanga and Manjakandriana, Madagascar between 2018 and 2019 (*P. rufus*: Ambakoana roost [−18.51 S, 48.17 E], Mahialambo roost [−18.11 S, 48.21 E], Mangarivotra roost [−18.28 S, 48.24 E], Marotsipohy roost [−18.39 S, 48.14 E]; *E. dupreanum*: Angavobe cave [−18.94 S, 47.95 E], Angavokely cave [−18.93 S, 47.76 E]; *R. madagascariensis*: Maromizaha cave [−18.96 S, 48.45 E]). Bats were live-captured using nets hung in tree canopies (*P. rufus*) and over cave mouths (*E. dupreanum, R. madagascariensis*) at dusk (17:00–22:00) and dawn (03:00–07:00). Captured bats were manually restrained, and bat sex, species, and age (juvenile or adult) were morphometrically determined in the field following previously published protocols [36,43–45]. Fecal and urine swabs were collected from all captured individuals, placed into viral transport medium, and frozen in liquid nitrogen. After sampling, swabs were transported to −80⁰C freezers at the Virology Unit at Institut Pasteur de Madagascar for long-term storage. In total, 1190 bats were captured (*P. rufus*: 129, *E. dupreanum*: 486, *R. madagascariensis*: 575).

This study was carried out in strict accordance with research permits obtained from the Madagascar Ministry of Forest and the Environment (permit numbers 019/18, 170/18, 007/19) and under guidelines posted by the American Veterinary Medical Association. All field protocols employed were pre-approved by the UC Berkeley Animal Care and Use Committee (ACUC Protocol #AUP-2017-10-10393), and every effort was made to minimize discomfort to animals.

## RNA extraction

A random subset of 285 fecal and 206 urine samples distributed across all three species was selected for downstream molecular analysis, including RNA extraction and mNGS (Feces – *P. rufus*: 26 male/18 female, *E. dupreanum*: 52 male/93 female, *R. madagascariensis*: 49 male/47 female; Urine – *P. rufus*: 48 male/47 female, *E. dupreanum*: 36 male/70 female, *R. madagascariensis*: 3 male/2 female). A subset of 24 urine samples were derived from prior fruit bat sampling carried out between 2013−2017 in the Districts of Moramanga, Mahabo, and Maroantsetra, Madagascar; see Madera et al. 2022 [38] for urine sampling details. RNA was extracted at the Virology Unit at the Institut Pasteur de Madagascar, using the Zymo Quick DNA/RNA Microprep Plus kit (Zymo Research, Irvine, CA, USA), adhering to the manufacturer's instructions while also including a DNAse digestion step. Water controls were extracted in conjunction with samples on each extraction day. Post-extraction, RNA underwent quality control on a nanodrop to assess its purity. A 260/280 ratio absorbance that did not exceed 2 was used to ensure that a quantifiable concentration was present. Extractions that passed screening were stored in freezers at −80$^0$C and transported to the Chan Zuckerberg Biohub (San Francisco, CA, USA) for library preparation and mNGS.

## Library preparation and mNGS

Four randomly selected samples from each bat species (*Pteropus rufus, Eidolon dupreanum, Rousettus madagascariensis*) underwent additional quantification using an Invitrogen Qubit 3.0 Fluorometer and the Qubit RNA HS Assay Kit (ThermoFisher Scientific, Carlsbad, CA, USA). Following quantification, RNA samples, and water samples from prior extraction, were pipetted into 96-well plates to automate high throughput mNGS library preparation. Based on initial quantification, 2uL aliquots from each plated sample were diluted 1:9 on a Bravo liquid handling platform (Agilent, Santa Clara, CA, USA). 5uL aliquots from each diluted sample were pipetted into 384-well plates for mNGS library preparation. Fecal samples were arrayed on distinct 384 well plates for sequencing runs. Each 384-well plate included additional RNA samples isolated from cultured HeLa cells and lab water samples, which served as controls for library preparation. Samples were transferred into a GeneVac EV-2 (SP Industries, Warminster, PA, USA) to conduct miniaturized mNGS library preparation with the NEBNext Ultra II RNA Library Prep Kit (New England BioLabs, Beverly, MA, USA). Library preparation was performed following manufacturer's instructions with a few modifications: 25pg of External RNA Controls Consortium Spike-in mix (ERCCS, Thermo-Fisher) was added to each sample prior to RNA fragmentation, input RNA mixture was fragmented for 8 minutes at 94$^0$C prior to reverse transcription, and a total of 14 cycles of PCR with dual-indexed TruSeq adapters was applied to amplify the resulting individual libraries. Resulting library pools then underwent quality and quantity measurements via electrophoresis (High-Sensitivity DNA Kit and Agilent Bioanalyzer; Agilent Technologies, Santa Clara, CA, USA), real-time quantitative polymerase chain reaction (qPCR) (KAPA Library Quantification Kit; Kapa Biosystems, Wilmington, MA, USA), and small-scale sequencing (2x146bp) on an iSeq platform (Illumina, San Diego, CA, US). Equimolar pooling of individual libraries from each plate (fecal and urine samples were sequenced on separate plates) was performed before running large-scale paired-end sequencing (2x146bp) on an Illumina NovaSeq sequencing system (Illumina, San Diego, CA, USA). Our fecal plate sequencing run totaled out at 10,493,788,488 short reads. The average number of pre-filtered short reads per sample was 34,747,644, while the average number of short reads that passed filtering per sample was 3,798,820. Our urine plate sequencing run totaled out at 4,009,622,408 short reads. The average number of pre-filtered reads per sample was 19,093,440, while the average number of short reads that passed filtering per sample was 2,665,585. The pipeline used to separate individual library outputs into FASTQ files of 146 bp paired-end reads can be found at https://github.com/czbiohub-sf/utilities.

## Prevalence of Kobuvirus sequence detection in field specimens

Raw reads recovered from Illumina sequencing were host-filtered, quality-filtered, and assembled on CZID (v3.10, NR/NT 2019-12-01), an open-source, cloud-based *de novo* assembly pipeline for microbial mNGS data [46], using publicly

available full-length bat genomes from GenBank at the time of sequencing (July 2019) as the host background model. Samples were deemed kobuvirus positive if CZID assembled at least two contigs with an average read depth >2 reads/nt that showed significant nucleotide or protein BLAST alignments (alignment length >100 nt/aa and E-value<0.00001 for nucleotide BLAST/ bit score >100 for protein BLAST) to kobuvirus reference sequences contained within NCBI NR/NT databases. As an exception, we also included single contigs that CZID annotated as kobuvirus 'positive' which exceeded 1,000 bp in length.

Offline BLASTn/x analyses of non-host contigs were conducted to cross-validate our search with CZID using a custom database of kobuvirus sequences from NCBI (last accessed: October 2021). Prior to BLAST searches, contigs were first deduplicated to remove redundant sequences using CD-HIT (v.4.8.1) [47]. BLAST hits from both searches identified sixteen contigs. One of these, a novel, full-genome length sequence that we eventually submitted to GenBank under accession number OP287812, was used as a reference for a third and final search aimed at filtering out low-quality hits from our prior runs. This high-quality contig was used as the representative sequence for downstream analyses.

### Genome quality assessment and annotation

We used CheckV (v1.0.1) [48] to estimate genome completeness and potential host contamination of our putative genomes. OP287812, one of our most complete contigs, was visualized in Geneious Prime (V.2023.0.1) and aligned to previously annotated kobuvirus sequences obtained from NCBI with MAFFT (v.1.5.0) [49]. Protease cleavage sites were identified and used to define individual proteins, with NCBI sequences serving as references. We assumed that the 5' and 3' ends encompassed regions flanking the single open reading frame. We annotated the rest of our kobuvirus-positive contigs using OP287812 as a reference sequence.

### Sequence similarity search

We conducted BLASTn and BLASTx searches to identify similarities between OP287812 and NCBI's database of kobuvirus sequences within Geneious. We organized BLAST hits using Geneious Prime's grade metric, a measure that produces a weighted score for hits composed of E-value, pairwise identity, and coverage to create a list of top 10 BLAST hits. We also generated an alignment between our kobuvirus-positive sequences and previously described bat kobuviruses, which we summarized using NCBI MSA Viewer (v.1.25.0).

### Genome coverage and sequence similarity analyses

We quantified read support for OP287812, which recovered a total of 31,282,224 pre-filtered short reads, from which 4,955,139 passed filters in CZID; using the raw read total, we calculated reads per million coverage across the full-length genome of OP287812. We next generated nucleotide and amino acid similarity plots comparing OP287812 to publicly available kobuvirus sequences recovered from NCBI (Accessions: KJ934637 and NC_001918). We pairwise-aligned these three sequences using the default parameter values in MAFFT, and supplied the following alignment as input to PySimPlot (v.0.1.1) [50] using default window (Default: 100) and step sizes (Default: 1). Further data analyses and visualizations were carried out in RStudio (v.2024.04.2+764) [51] using the tidyverse suite (v.2.0.0) [52].

### Phylogenetic analysis

We integrated our novel sequences with publicly available sequences on NCBI to build three separate phylogenies: (a) a picornavirus maximum-likelihood (ML) tree spanning a conserved 7,000 bp region, (b) a kobuvirus-only ML tree spanning a conserved 4,957 bp region, and (c) a time-resolved Bayesian kobuvirus-only phylogeny spanning a conserved 5,500 bp region. Sequences were aligned with MAFFT under default parameters and subjected to ModelTest-NG (v.0.1.7) [53] to determine the best fit nucleotide substitution models to describe evolutionary relationships within each respective alignment. ML trees

were built using RAxML (v.8.2.13) [54] and visualized within RStudio using the ggtree (v3.16) [55] package. Following standard practice outlined in the RAxML-NG manual, we computed 20 tree searches using 10 random and 10 parsimony-based starting trees under default heuristic search parameters for each original alignment, then selected the best-scoring topology. MRE-based bootstrapping tests were performed after every 50 replicates [56], following Felsenstein's method [57], terminating at 1,000 bootstrap replicates. A similar approach was used to construct our Bayesian phylogenetic tree with BEAST2 (v2.6.3) [58,59], with the key difference being that representative sequences were selected from across the Kobuvirus-only ML phylogeny using Parnas (v.0.1.4) [60] to ensure adequate coverage of tree diversity. More details for the generation of each phylogeny are available in our open-access GitHub repository (see Data Availability).

## Results

### Kobuvirus detected by mNGS at moderate prevalence in samples derived from *E. dupreanum*

Fifteen (15/285) fecal samples originating from different individual *Eidolon dupreanum* bats were kobuvirus positive via offline BLAST analyses or CZID filtering (5.26% positivity) (**Table 1**). One additional kobuvirus sequence was detected in a single adult female *E. dupreanum* urine sample collected from Angavokely roost in April 2019 and was also included in this study (OQ818322). Urine swabs were collected only opportunistically when urination was witnessed in the field, resulting in frequent contact with the bats' anal region; as such, we hypothesize this urine-derived kobuvirus sequence to likely result from contamination with adjacent fecal material. No paired fecal sample was available for sequence comparison from this individual bat; however, the sequence displayed strong genomic and phylogenetic similarity to all other fecal-derived contigs. Samples collected from *P. rufus* and *R. madagascariensis* did not demonstrate any evidence of kobuvirus carriage.

Our most complete fecal-derived kobuvirus genome (OP287812) was collected from a juvenile *E. dupreanum* female in Angavokely cave in July 2018. Sequence OP287812 is 8,263 bp in length and was designated as 'high quality' by CheckV (completeness = 100, contamination = 0, CheckV quality = High quality, MIUVIG Quality = High quality). Additional contig quality assessments and associated metadata for our other identified genomes can be found in S1 Table and within our GitHub repository. These sequences represent the most complete bat kobuvirus genomes identified to date and the first bat kobuviruses to be identified in Madagascar.

### Genome annotation and comparative genomic analysis of the full-length kobuvirus genome OP287812 reveals homology across clades

We annotated the single ORF that spans OP287812 (7,305 nt and 2,435 aa) and identified protease cleavage sites across the genome (**Fig 1A**). Reference sequences used to annotate the OP287812 genome can be found in S2 Table. Predicted cleavage sites occurring at junctions between L and VP0 (Glutamine/Glycine), 2A and 2B (Glutamine/Glycine), 2B and 2C

**Table 1. Positive Kobuvirus Samples.**

| Roost site | Species | Total sampled by site (n = 285) | Total Kobuvirus positive (n = 15) | Total sampled (Male, Female) | Total Kobuvirus positive (Male, Female) |
|---|---|---|---|---|---|
| Ambakoana | *Pteropus rufus* | 37 | 0 (0%) | 23,14 | 0,0 (0%, 0%) |
| Angavobe | *Eidolon dupreanum* | 37 | 10 (27.0%) | 11,26 | 2,8 (18.18%, 30.77%) |
| Angavokely | *Eidolon dupreanum* | 108 | 5 (4.63%) | 41,67 | 4,1 (9.75%, 1.49%) |
| Maromizaha | *Rousettus madagascariensis* | 96 | 0 (0%) | 49,47 | 0,0 (0%, 0%) |
| Mahialambo | *Pteropus rufus* | 7 | 0 (0%) | 3,4 | 0,0 (0%, 0%) |

Summary table showing total bats captured by species and location from a random subset of fecal samples subject to mNGS. *Only one *E. dupreanum* urine sample out of 206 total urine samples recovered from all three bat species was kobuvirus-positive. See Madera et al. 2022 [38] for detailed breakdown of bat captures from which urine samples were recovered.

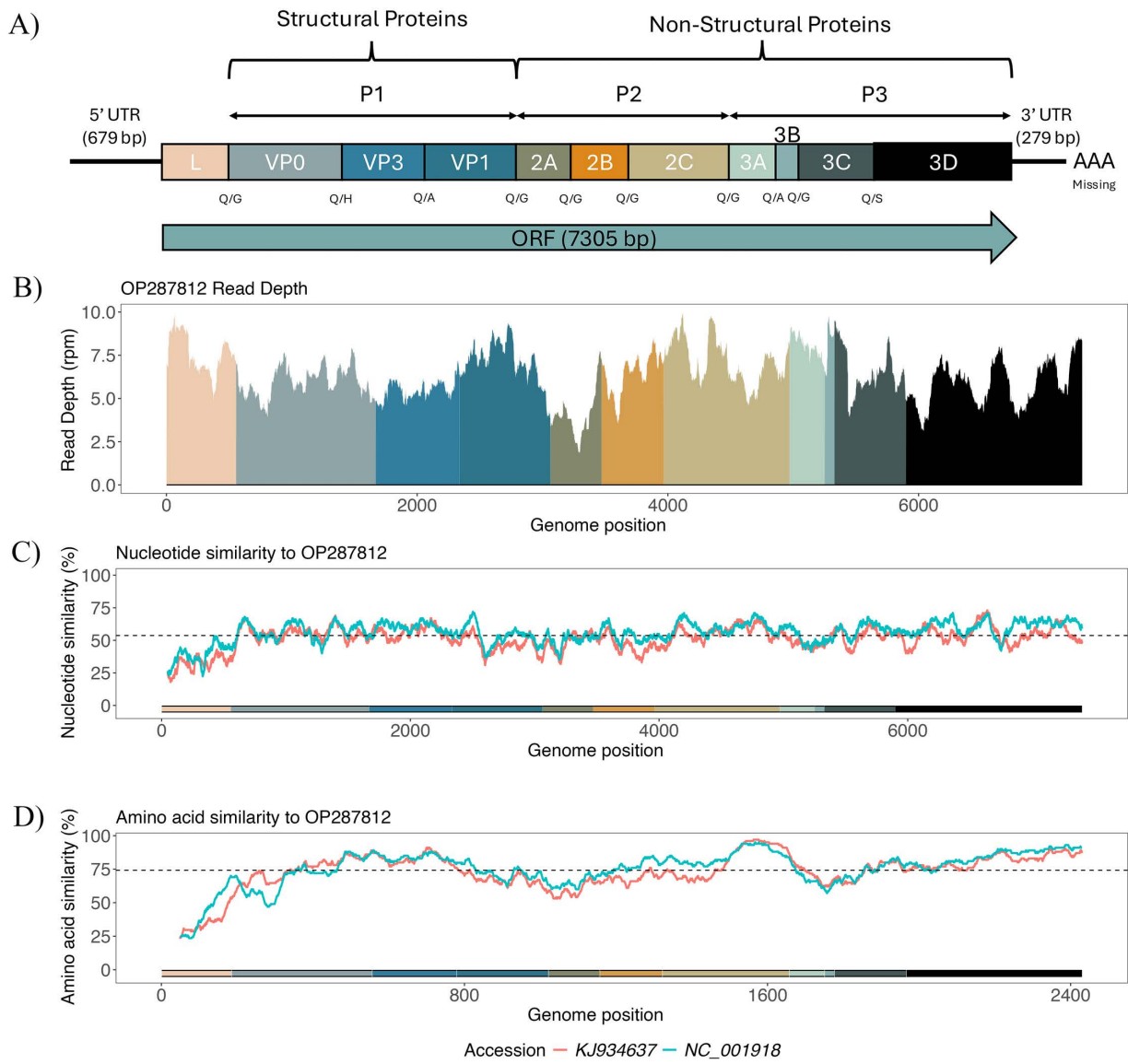

**Fig 1. Genome Annotation and Similarity plot for OP287812. A)** Genome annotation for OP287812 is shown at the top, with amino acid cleavage sites indicated below their corresponding proteins. Proteins are color-coded and displayed in panels **B-D. B)** Coverage map across OP287812's open reading frame (ORF), excluding the 5' and 3' UTRs. Sequence depth is represented in reads per million (rpm), with colors indicating the proteins along the ORF. **C)** and **D)** Nucleotide and amino acid similarity plots, with OP287812 as the reference. The plots compare OP287812 to the most derived human kobuvirus sequence (NC_001918), and the most basal avian kobuvirus (KJ934637), in subsequent phylogenetic trees. Dashed lines indicate the average percent similarity between OP287812 and the query sequences.

(Glutamine/Glycine), 3A and 3B (Glutamine/Serine), and 3C and 3D (Glutamine/Serine) were consistent with prior findings in kobuviruses carried by other hosts [61,62], highlighting the genomic conservation of kobuviruses across diverse host species.

We then plotted genome similarity between OP287812 and the most basal Aichivirus A variant, as well as between OP287812 and the most derived Aichivirus A variant included in our phylogenies (Fig 1C–1D). We found that OP287812 shares an average nucleotide identity of 53.70% (**Fig 1C**) and an average amino acid sequence similarity of 74.19% (**Fig**

1D) to previously described kobuviruses (NC_001918 – Human and KJ934637 – Avian), consistent with BLAST results in which we recover high genome similarity to Aichivirus A variants.

We conducted additional BLAST searches of OP287812 against publicly available sequences in NCBI to assess its genomic similarity to previously identified kobuviruses (Table 2). Whole-genome BLASTn searches revealed a top hit to a human kobuvirus (Accession: GQ927711) covering 87.23% of the query and demonstrating 74.10% pairwise identity to OP287812. Additionally, one BLASTn hit indicated homology to a partial *Eidolon helvum* kobuvirus sequence from Ghana, which resolved as basal to canid and human Aichivirus A in phylogenetic analysis (Accession: JX885611, Peptide: L Peptide, Contig Length: 1,120 bp, Query Coverage: 100%, Pairwise Identity: 96.60%) [30]. Most hits indicated homology to Aichivirus A variants and coincided with findings from our BLASTx search (S3 Table).

**Phylogenetic analysis suggests common ancestry between bat kobuvirus OP287812 and Aichivirus A genotypes**

We generated a nucleotide-based ML phylogeny for the *Picornaviridae* family, incorporating our newly identified *E. dupreanum* sequence (OP287812) alongside Picornavirus sequences obtained from NCBI (Fig 2). OP287812 clustered within the previously described kobuviruses, consistent with earlier BLAST analyses. Specifically, OP287812 localized within the Aichivirus A subclade of kobuvirus. This clade received strong support, with a bootstrap value of 100/100. Notably, previously described bat kobuvirus sequences (Accessions: KJ641691 and KJ641686) [29] did not cluster with our Malagasy bat kobuvirus. We also constructed a second ML phylogeny focusing specifically on sequences within the kobuvirus genus. This tree included previously identified bat kobuvirus sequences visualized in our Picornaviridae tree, bat kobuvirus sequences from Vietnam [12], and 10 near-complete kobuvirus sequences recovered from *E. dupreanum* in our study (S1 Fig). All *E. dupreanum* sequences clustered closely with OP287812. Vietnamese bat sequences also clustered within the Aichivirus A subclade but grouped separately and were more derived compared to our E. dupreanum sequences. Alignment statistics between our novel Malagasy sequences and these previously identified bat kobuviruses can be found in S4 Table.

**Table 2. Top BLASTn Hits for OP287812.**

| Genome Region | Start (nt) | End (nt) | Predicted Protease Cleavage Site | Pairwise Identity % | E-Value | NCBI Accession | Hit Length (bp) |
|---|---|---|---|---|---|---|---|
| Whole Genome | 1 | 8263 | – | 74.10% | 0 | GQ927711 | 7240 |
| 5'UTR | 1 | 679 | – | 70.70% | 8.88E-60 | MK671314 | 657 |
| ORF | 680 | 7984 | – | 74.70% | 0 | MW292482 | 7016 |
| L | 680 | 1234 | Q/G | 96.60% | 0 | JX885611 | 554 |
| VP0 | 1235 | 2347 | Q/H | 77.10% | 0 | MN648601 | 1117 |
| VP3 | 2348 | 3016 | Q/A | 78.60% | 2.50E-155 | MK201778 | 668 |
| VP1 | 3017 | 3742 | Q/G | 72.20% | 5.28E-101 | KJ950958 | 724 |
| 2A | 3743 | 4150 | Q/G | 74.80% | 2.85E-63 | KJ950958 | 391 |
| 2B | 4151 | 4645 | Q/G | 76.10% | 8.83E-90 | MF175074 | 478 |
| 2C | 4646 | 5650 | Q/G | 78.10% | 0 | MG200054 | 995 |
| 3A | 5651 | 5929 | Q/A | 68.40% | 1.86E-08 | KJ934637 | 209 |
| 3B | 5930 | 6007 | Q/G | 78.30% | 3.94E-06 | MT610361 | 80 |
| 3C | 6008 | 6577 | Q/S | 76.00% | 8.40E-110 | GQ927706 | 569 |
| 3D | 6578 | 7984 | – | 81.10% | 0 | MH052678 | 1406 |
| 3'UTR | 7985 | 8263 | – | – | – | – | – |

Identity percentages, E-Values, corresponding hit lengths, and NCBI Accessions for the highest-ranking BLASTn hit between OP287812 and NCBI kobuvirus sequences. Nucleotide lengths for the whole OP287812 genome, including the ORF, individual proteins, and the UTRs, along with the predicted protease cleavage sites (in single-letter amino acid code) marking the start and end positions of each protein, are included. No hits were observed for the 3'UTR.

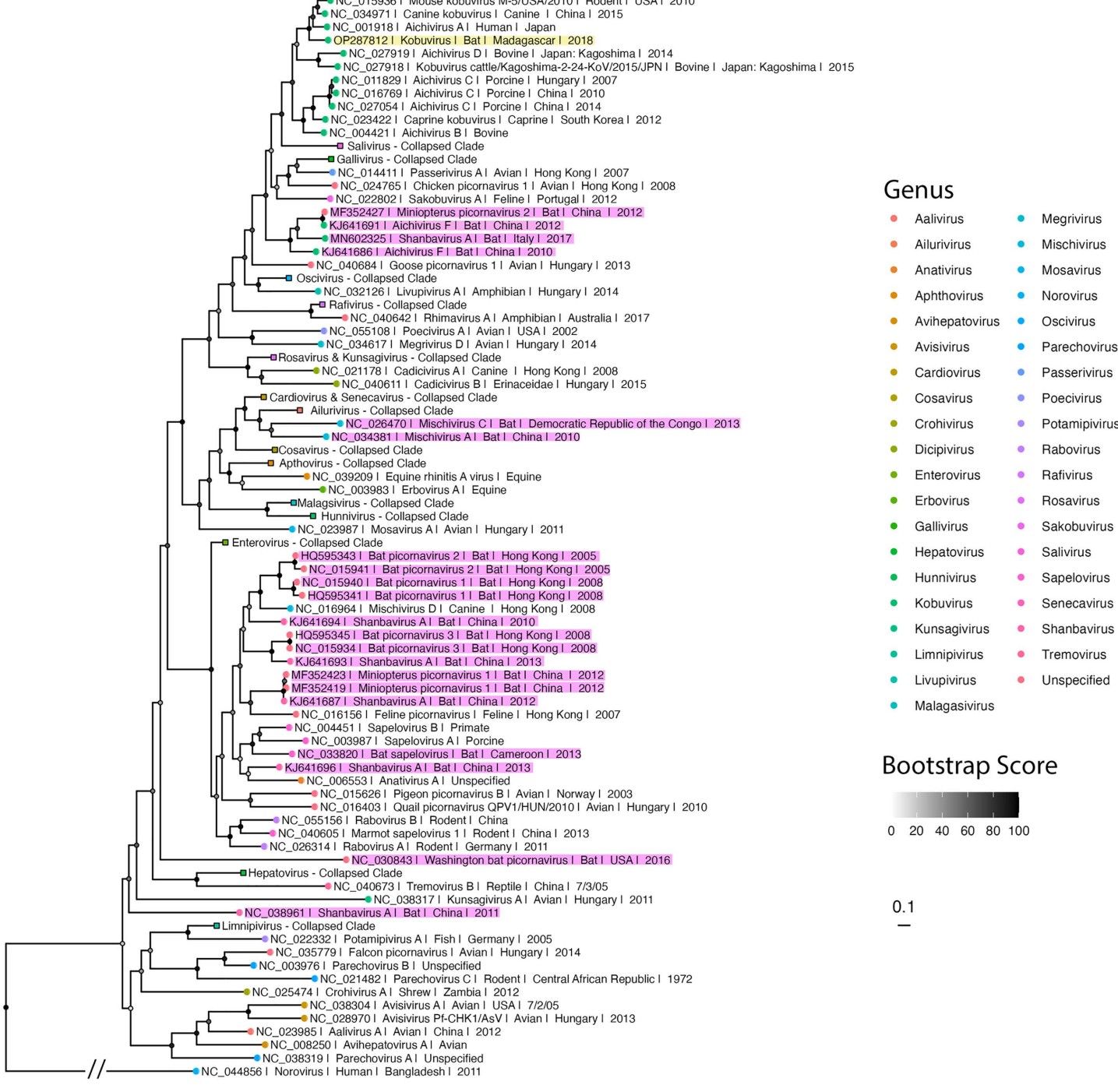

**Fig 2. Phylogenetic analysis of OP287812 among previously identified picornaviruses.** Maximum likelihood phylogeny of a conserved 7,000 bp region of *Picornaviridae* sequences (nucleotide substitution model: TVM + I + G4). Node color, represented in greyscale, indicates bootstrap support, with darker shades corresponding to higher support values and lighter shades to lower support values. OP287812 is highlighted in yellow, while bat *Picornaviridae*, excluding OP287812, are highlighted in pink. Tip points are colored by broad viral taxonomic groups. Tip labels include NCBI accession number, virus species or genus, host, geographic origin, and year of identification, as available from NCBI. Branch lengths are scaled by nucleotide substitutions per site, noted by the scalebar. The tree is rooted with human norovirus (NC_044856). The branch length of this outgroup was shortened to improve phylogenetic tree visualization and is denoted as such with a double hash.

Finally, our time tree (**Fig 3**) estimated the most recent common ancestor (MRCA) for all kobuviruses to be in the year 1396 (~628 years ago; 95% HPD: 1228–1556). It also supported the clustering of our Malagasy kobuvirus among Aichivirus A variants, with OP287812 diverging from its closest relatives at an MRCA dated to 1882 (~142 years ago; 95% HPD: 1846–1914). This divergence occurred after an ancestral avian variant (GenBank Accession: KJ934637) diverged

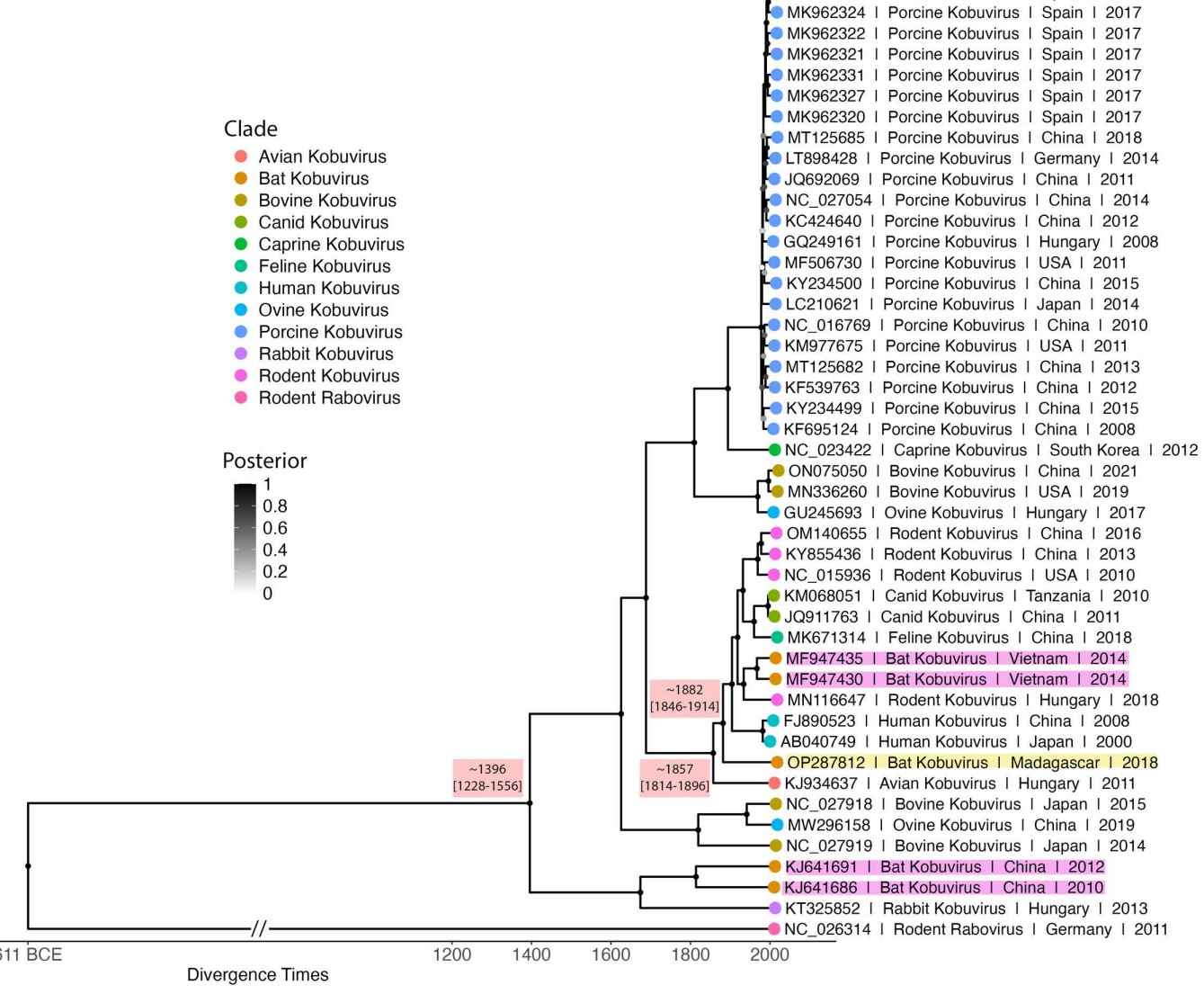

**Fig 3. Bayesian phylogeny estimating time to MRCA for our novel *Eidolon dupreanum* Kobuvirus (OP287812) and other kobuvirus sequences.** The Bayesian tree was constructed with 700 million runs of a strict molecular clock Bayesian Skyline Coalescent model (GTR+I+G4), implemented in BEAST2, on an alignment of a conserved 5,500 bp region of the kobuvirus genome. Node color, represented in greyscale, indicates posterior support, with darker shades corresponding to higher support values and lighter shades to lower support values after averaging of all 700 million trees after 10% burn-in. OP287812 is highlighted in yellow, while other bat kobuviruses are highlighted in pink. Tip points are colored by kobuvirus clade. Tip labels include NCBI accession number, clade, geographic origin, and year of identification, as available from NCBI. Estimated divergence times, with 95% highest posterior density (HPD) intervals, are depicted alongside key nodes. The outgroup is rodent rabovirus (NC_026314). The branch length of this outgroup was shortened to improve phylogenetic tree visualization and is denoted as such with a double hash.

from the lineage approximately in 1857 (~25 years prior; 95% HPD: 1814–1896). These divergence estimates suggest a relatively recent evolutionary origin for kobuviruses.

## Discussion

Characterizing virus diversity in wildlife hosts serves as a foundational step towards downstream comprehensive analysis of host-virus ecology, transmission dynamics, and zoonotic risk. Bats are critical targets for pathogen surveillance due to their role as reservoirs for numerous zoonotic viruses, including coronaviruses, filoviruses, and henipaviruses. Effective surveillance in bat populations not only aids in detecting novel pathogens before they spill over into human populations but also provides key insights into the ecological and evolutionary factors that influence virus maintenance and transmission within and between species.

We found that Malagasy bat kobuviruses phylogenetically cluster with Aichivirus A genotypes, alongside previously described bat kobuviruses and other notable mammalian kobuviruses including those found in humans, birds, felids, canids, and rodents [12]. Malagasy bat kobuviruses appear basal to the vast mammalian host radiation [12] that has since taken place within the Aichivirus A subclade. Our study highlights the growing complexity of kobuvirus classification, suggesting that bat kobuviruses form a paraphyletic group across Aichivirus A and Aichivirus F clades, interspersed with sequences derived from a variety of other vertebrate hosts, including humans [12,63]. This highlights kobuviruses' substantial potential for cross-species transmission, including zoonosis. Prior studies document robust evidence of cross-species kobuvirus transmission from bats to rabbits and more muted evidence of kobuvirus transmission from bats to rodents [12]. Additionally, compelling evidence for kobuvirus transmission between other vertebrate hosts has been identified on farms where a novel kobuvirus sequenced from sheep in paddocks near cattle farms nested phylogenetically within the bovine kobuvirus clade, suggesting transmission from bovine to ovine hosts [64,65]. As the evidence for kobuvirus circulation across diverse animal hosts continues to grow, continued kobuvirus surveillance in bat populations will be essential to identifying potential spillover events early. Bat proximity to humans and livestock increases the risk of viral spillover [42], and while bat kobuviruses have not yet been linked to major disease outbreaks, their capacity for cross-species transmission presents a serious emerging zoonotic threat.

Given these concerns, comprehensive investigations into the transmission mechanisms of kobuviruses in wildlife are critical. Kobuviruses are presumed to be spread via fecal-oral transmission or environmental contamination, but it remains unclear whether they can also be transmitted through breastfeeding or as blood-borne pathogens [66]. Their replication mechanisms and host cell receptors are similarly poorly understood, though comparisons between kobuviruses and related enteroviruses offer clues. For example, in poliovirus and coxsackievirus, transmission of virus between cells exposes the N-terminal region of the VP1 capsid protein, subsequently releasing the VP4 capsid protein [67–70]; this process is absent in kobuviruses which lack VP4 in their capsid. These topological differences suggest kobuviruses may use distinct host cell receptor binding sites compared to other *Picornaviridae.* Phylogenetic analyses of capsid proteins and RdRp further suggest that these proteins have evolved separately within kobuviruses [67,71]. While the sources of these variations remains unclear, recombination – which drives genome modularity and evolution in RNA viruses broadly [72] – could play a role. The high genomic similarity across the kobuvirus genus and among diverse host species suggests that recombination may drive cross-species transmission and virus diversification [73]. Still, the role of recombination and other factors shaping kobuvirus host range remains to be determined.

A clear geographical bias exists in bat kobuvirus sampling, as all previously described bat kobuvirus sequences used in our analyses (MF947429-MF947440 – *Scotophilus kuhlii*, KJ641691 – *Miniopterus fulginosus,* KJ641686 – *Myotis ricketti*) are exclusively derived from Asian bat species [12,29]. We note that BLAST analyses indicated high similarity between OP287812 and an 1,120 bp kobuvirus sequence (Accession Number: JX885611) recovered from *E. helvum* in Ghana [30]. This *E. helvum* sequence was too short for inclusion in our *Picornaviridae* or kobuvirus phylogenies; however, if recovered at full length, its BLASTn similarity suggests that the two may resolve as a monophyletic *Pteropodid* clade for kobuviruses

distinct from those recovered from insectivorous bats in Asia. Expanding surveillance across diverse regions will be essential to bridging these gaps. Madagascar's unique dual African-Asian evolutionary history may hold key insights into ancestral bat kobuviruses and *Picornaviridae* more broadly. Our comparative analyses reveal conservation in genomic regions responsible for viral entry and replication [63,74,75], such as VP1 and RdRP, between Madagascar bat kobuvirus OP287812 and kobuviruses previously identified elsewhere. Future studies should leverage existing PCR protocols targeting these conserved regions, like RdRp [74,76], to facilitate long-term kobuvirus surveillance in resource-limiting settings. These would enable tracking of viral dynamics while refining our understanding of kobuvirus geographic distribution and potential spillover risks – particularly critical in Madagascar, where human-wildlife interactions may further influence viral transmission dynamics.

Madagascar presents a unique environment for virus diversification due to its long isolation, dual African-Asian phylogeographic history, and high endemicity of unique mammalian fauna. Among these factors, wild game consumption plays a particularly complex role in shaping human-wildlife interactions and potential viral spillover. Wild game trade and consumption patterns in Madagascar vary widely by region and community, offering an important nutritional and cultural resource [77,78]. Rural communities often hunt for sustenance or to protect crops from frugivorous bats, whereas urban communities primarily consume wild game for taste preferences [77,79–81]. Despite some wildlife populations being protected due to cultural reasons [82–84], unsustainable and illegal hunting practices [77,79,85], along with habitat destruction, continue to threaten wildlife. Anthropogenic threats can drive bats into synanthropic lifestyles, increasing the risk of zoonotic disease transmission [42,82,86]. Conservation efforts may thus offer benefits in terms of zoonotic disease reduction, but any proposed changes to hunting laws must also address human reliance on wildlife for nutrition [77,78,82,87–89].

## Conclusions

mNGS-based surveillance for bat viruses has resulted in incredibly diverse datasets, allowing for novel insights into wild bat virus ecology and evolution [29,38–40,90,91]. Here, we expand the known host and geographic range of kobuviruses to include *E. dupreanum* fruit bats of Madagascar. We describe the most complete bat kobuvirus genomes identified to date, offering a glimpse into the origin and diversification of the kobuvirus genus more broadly. We find that bat kobuviruses in Madagascar phylogenetically nest among Aichivirus A genotypes and are highly divergent from previously described bat kobuviruses. Genome similarity analyses demonstrate significant conservation of kobuvirus genomic content across clades, particularly in regions of the virus genome involved in virus entry and replication. Further analyses are needed to determine whether this trend holds for other kobuviruses obtained from bats in various geographic landscapes, including unsampled species within Madagascar, or if the high identity between bat- and human-hosted kobuviruses identified here is unique to the region. While we did not identify kobuviruses in other sampled species (*P. rufus* and *R. madagascariensis*), these species should remain a focus of future sampling efforts, as their inclusion is essential for understanding the full ecological and evolutionary dynamics of bat kobuviruses. Moreover, our discovery of several partial kobuvirus genomes in multiple *E. dupreanum* individuals highlights the need for more comprehensive sampling. Given the high rates of human-bat contact in Madagascar, our findings raise concern for public health. We strongly advocate for enhanced surveillance and detection efforts to further elucidate the ecology of these viruses in their wild bat hosts, as well as other animal hosts, as these efforts are critical for understanding their potential impact on both wildlife conservation and zoonotic disease transmission.

## Supporting information

**S1 Fig. Phylogenetic analysis of OP287812 among previously identified kobuviruses.** Maximum likelihood phylogeny of kobuvirus sequences (nucleotide substitution model: GTR+I+G4). Node color, represented in greyscale, indicates bootstrap support, with darker shades corresponding to higher support values and lighter shades to lower support values. Madagascar bat kobuvirus sequences are highlighted in yellow, while other bat kobuviruses are highlighted in pink. Tip

points are colored by kobuvirus clade. Tip labels include NCBI accession number, virus species, clade, geographic origin, and year of identification, as available from NCBI. Branch lengths are scaled by nucleotide substitution per site, noted by the scalebar. The tree is rooted with rodent rabovirus (NC_026314). The branch length of this outgroup was shortened to improve phylogenetic tree visualization and is denoted as such with a double hash.
(TIF)

**S1 Table. Kobuvirus-positive contig quality assessment and corresponding metadata.** Summary of minimum information about uncultivated virus genomes (MIUViG) per CheckV and corresponding metadata for all contigs identified as positive kobuvirus hits. Age statuses: NL = Non-Lactating Female, J = Juvenile, A = Adult, P = Pregnant. Accession OQ818322 is marked with an asterisk (*) to note that is derived from urine.
(DOCX)

**S2 Table. Reference Sequences Used for Annotating the OP287812 Genome.** Summary of reference sequences used to annotate the OP287812 genome and their corresponding NCBI accession number, including the percent similarity between the reference sequence and OP287812. Columns without data indicate regions that were manually annotated (i.e., 5' UTR, L Peptide, 3' UTR) without the use of a reference sequence.
(DOCX)

**S3 Table. Top BLASTx Hit for OP287812.** Identity percentages, E-Values, corresponding hit lengths, and NCBI Accessions for the highest-ranking BLASTx hit between OP287812 and NCBI kobuvirus sequences. Amino acid lengths for the OP287812 genome, as well as predicted protease cleavage sites (in single-letter amino acid code) indicating start and end positions for individual proteins, are listed. No hits were observed within the 5'UTR and 3'UTR.
(DOCX)

**S4 Table. Genome alignment statistics between novel Madagascar bat Kobuviruses and previously identified bat kobuviruses with OP287812 set as the reference sequence.** Alignment statistics between bat kobuvirus sequences used in **Supplemental Figure 1**. Accession OQ818322 is marked with an asterisk (*) to note that is derived from urine.
(DOCX)

## Acknowledgments

We thank Anecia Gentles, Kimberly Rivera, Fifi Ravelomanantsoa, and Sarah Guth for help in the field and lab. We acknowledge the Virology Unit at the Institut Pasteur de Madagascar for logistical support, and we thank the Mention of Zoology and Animal Biodiversity at the University of Antananarivo and the Madagascar Ministry of the Environment and Sustainable Development for providing research and export permits. We thank Amy Kistler, Vida Ahyong, Angela Detweiler, Michelle Tan, and Norma Neff of the Chan Zuckerberg Biohub (CZB) for sequencing support and Cristina M. Tato, Maira Phelps, and Joseph L. DeRisi of CZB for logistical support. We thank the Brook lab at the University of Chicago for helpful contributions to the manuscript. This work was completed in part with resources provided by the University of Chicago's Research Computing Center.

## Author contributions

**Conceptualization:** Freddy Leonel Gonzalez, Cara E. Brook.

**Data curation:** Freddy Leonel Gonzalez.

**Formal analysis:** Freddy Leonel Gonzalez.

**Funding acquisition:** Freddy Leonel Gonzalez, Cara E. Brook.

**Investigation:** Freddy Leonel Gonzalez.

**Methodology:** Freddy Leonel Gonzalez.

**Project administration:** Freddy Leonel Gonzalez, Cara E. Brook.

**Resources:** Freddy Leonel Gonzalez, Cara E. Brook.

**Software:** Freddy Leonel Gonzalez.

**Supervision:** Cara E. Brook.

**Validation:** Freddy Leonel Gonzalez, Cara E. Brook.

**Visualization:** Freddy Leonel Gonzalez.

**Writing – original draft:** Freddy Leonel Gonzalez.

**Writing – review & editing:** Freddy Leonel Gonzalez, Gwenddolen Kettenburg, Hafaliana Christian Ranaivoson, Angelo Andrianiaina, Santino Andry, Vololoniaina Raharinosy, Tsiry Hasina Randriambolamanantsoa, Vincent Lacoste, Philippe Dussart, Jean-Michel Héraud, Cara E. Brook.

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
