## [Decision Letter · Decision Letter 0]

25 Feb 2025

PONE-D-25-01022Genomic characterization of novel bat kobuviruses in Madagascar: implications for viral evolution and zoonotic riskPLOS ONE

Dear Dr. Gonzalez,

Thank you for submitting your manuscript to PLOS ONE. After careful consideration, we feel that it has merit but does not fully meet PLOS ONE’s publication criteria as it currently stands. Therefore, we invite you to submit a revised version of the manuscript that addresses the points raised during the review process.

Please submit your revised manuscript by Apr 11 2025 11:59PM. If you will need more time than this to complete your revisions, please reply to this message or contact the journal office at plosone@plos.org . Please include the following items when submitting your revised manuscript:

We look forward to receiving your revised manuscript.

Kind regards,

Joshua Kamani, PhD

Academic Editor

PLOS ONE

“Funding: This work was funded by the National Institutes of Health (1R01AI129822-01 grant to J-MH, PD, and CEB and 5DP2AI171120 grant to CEB), DARPA (PREEMPT Program Cooperative Agreement no. D18AC00031 to CEB), the Bill and Melinda Gates Foundation (GCE/ID OPP1211841 to CEB and J-MH), the Adolph C. and Mary Sprague Miller Institute for Basic Research in Science (postdoctoral fellowship to CEB), the Branco Weiss Society in Science (fellowship to CEB), the Chan Zuckerberg Biohub, and the University of Chicago PREP program (5R25GM066522 grant, fellowship to FLG).”

Additional Editor Comments:

Authors should mention in the introduction or discussion if there are social, cultural or anthropogenic activities peculiar to Madagascar that encourages human-bat contact and constitute public health risks.

Reviewers' comments:

Reviewer's Responses to Questions

**Comments to the Author**

1. Is the manuscript technically sound, and do the data support the conclusions?

Reviewer #1: Yes

2. Has the statistical analysis been performed appropriately and rigorously? 

Reviewer #1: Yes

3. Have the authors made all data underlying the findings in their manuscript fully available?

Reviewer #1: Yes

4. Is the manuscript presented in an intelligible fashion and written in standard English?

Reviewer #1: Yes

5. Review Comments to the Author

Reviewer #1: This manuscript makes a valuable contribution to understanding the diversity of kobuviruses in wildlife, particularly bats in Madagascar. It is well-organized and the findings are significant in the field of cross-species transmission of novel viruses. I recommend to strengthen the manuscript further particularly the broader implications of the findings for zoonotic risk.

1. In the Introduction section, it would be valuable to discuss the specific ecological or biological factors that make kobuviruses in bats even if other wildlife species more prone to cross-species transmission.

2. The authors undertook the metagenomic Next Generation Sequencing on stool samples from Madagascar fruit bats to feature potential zoonotic viruses. More information about the sequencing depth and any steps taken to ensure data quality should be included.

3. It is important to discuss the specific mechanisms of cross-species transmission for kobuviruses between different wildlife species.

6. PLOS authors have the option to publish the peer review history of their article (what does this mean? ). If published, this will include your full peer review and any attached files.

**Do you want your identity to be public for this peer review?** For information about this choice, including consent withdrawal, please see our Privacy Policy .

Reviewer #1: No

---

## [Author Response · Author response to Decision Letter 1]

24 Jul 2025

All responses can be found in the 'Response to Reviewers' document

---

## [Editor Report · Decision Letter 1]

20 Aug 2025

Genomic characterization of novel bat kobuviruses in Madagascar: implications for viral evolution and zoonotic risk

PONE-D-25-01022R1

Dear Dr.

We’re pleased to inform you that your manuscript has been judged scientifically suitable for publication and will be formally accepted for publication once it meets all outstanding technical requirements.

Kind regards,

Joshua Kamani, PhD

Academic Editor

PLOS ONE
---

## [Editor Report · Acceptance letter]

PONE-D-25-01022R1

PLOS ONE

Dear Dr. Gonzalez,

I'm pleased to inform you that your manuscript has been deemed suitable for publication in PLOS ONE. Congratulations! Your manuscript is now being handed over to our production team.

Kind regards,

on behalf of

Dr. Joshua Kamani

Academic Editor

PLOS ONE